# Exploring the Roles of Key Mediators IKBKE and HSPA1A in Alzheimer’s Disease and Hepatocellular Carcinoma through Bioinformatics Analysis

**DOI:** 10.3390/ijms25136934

**Published:** 2024-06-25

**Authors:** Yiying Wang, Yakun Yang, Ce Liang, Hailin Zhang

**Affiliations:** Department of Pharmacology, The Key Laboratory of Neural and Vascular Biology, Ministry of Education, The Key Laboratory of New Drug Pharmacology and Toxicology, Collaborative Innovation Center of Hebei Province for Mechanism, Diagnosis and Treatment of Neuropsychiatric Diseases, Hebei Medical University, Shijiazhuang 050017, China; pharmacy_wyy@163.com (Y.W.); 19201608@hebmu.edu.cn (Y.Y.); 19201480@hebmu.edu.cn (C.L.)

**Keywords:** AD, LIHC, IKBKE, HSPA1A, bioinformatics analysis, machine learning

## Abstract

Recent studies have hinted at a potential link between Alzheimer’s Disease (AD) and cancer. Thus, our study focused on finding genes common to AD and Liver Hepatocellular Carcinoma (LIHC), assessing their promise as diagnostic indicators and guiding future treatment approaches for both conditions. Our research utilized a broad methodology, including differential gene expression analysis, Weighted Gene Co-expression Network Analysis (WGCNA), gene enrichment analysis, Receiver Operating Characteristic (ROC) curves, and Kaplan–Meier plots, supplemented with immunohistochemistry data from the Human Protein Atlas (HPA) and machine learning techniques, to identify critical genes and significant pathways shared between AD and LIHC. Through differential gene expression analysis, WGCNA, and machine learning methods, we identified nine key genes associated with AD, which served as entry points for LIHC analysis. Subsequent analyses revealed *IKBKE* and *HSPA1A* as shared pivotal genes in patients with AD and LIHC, suggesting these genes as potential targets for intervention in both conditions. Our study indicates that *IKBKE* and *HSPA1A* could influence the onset and progression of AD and LIHC by modulating the infiltration levels of immune cells. This lays a foundation for future research into targeted therapies based on their shared mechanisms.

## 1. Introduction

Alzheimer’s Disease (AD) is a progressive neurodegenerative disorder and the most common cause of dementia in the elderly. Its pathological hallmarks are the insidious accumulation of amyloid-β plaques and the presence of neurofibrillary tangles within the brain, leading to the gradual erosion of cognitive function and memory [1,2,3]. The etiology of AD is multifactorial, involving an intricate interplay between genetic, environmental, and lifestyle factors, ultimately leading to synaptic dysfunction and neuronal loss [4,5,6]. The molecular mechanisms associated with AD are yet to be fully elucidated, which has made the identification of robust biomarkers a contemporary research imperative.

Cancer is firmly established as one of the preeminent causes of mortality worldwide, posing a significant challenge to global health with its profound impact on individuals, families, and healthcare systems [7]. Some studies have pointed to a potential inverse correlation between cancer and AD, while others argue for a parallel relationship [8,9,10]. Although the exact mechanisms underlying the relationship between cancer and AD remain to be fully understood, both conditions share similar characteristics and risk factors, including aging, smoking, obesity, and Type 2 diabetes [11,12,13,14,15]. The pathogenesis of both AD and cancer involves the PI3K/AKT/mTOR signaling pathway, which is crucial for regulating cell proliferation, metabolism, growth, and autophagy [16,17,18,19,20]. Additionally, angiogenesis, inhibition of cell adhesion, and inflammation are common features in both cancer and AD [9]. Therefore, elucidating the commonalities between cancer and AD is of paramount importance, potentially unlocking novel therapeutic avenues and offering new hope for the treatment of these diseases.

In this research, we utilized bioinformatics and machine learning approaches to discern the links between AD and Liver Hepatocellular Carcinoma (LIHC). Bioinformatics facilitates the comprehensive analysis of gene expression datasets to identify potential biomarkers for disease diagnosis and prognosis [21,22]. Machine learning explores pathological mechanisms and therapeutic targets at the genetic level across different diseases [23]. Through a combined bioinformatics and machine learning analysis, we have identified key genes and molecular mechanisms shared between AD and LIHC, offering significant insights for a potential cure for these two diseases.

## 2. Results

### 2.1. Identification of Differentially Expressed Genes (DEGs) in AD

The flow chart of this study is shown in Figure 1. Initially, we normalized the data and illustrated the distribution of the normalized expression values using a box plot (Figure 2A). This step ensured the quality and comparability of the dataset. Following normalization, we conducted a differential gene expression analysis on the GSE132903 dataset, identifying a total of 14,677 differentially expressed genes (DEGs). Among these, 7436 genes were upregulated and 7241 genes were downregulated in the Alzheimer’s disease (AD) samples compared to the control samples. The DEGs were visualized using a volcano plot (Figure 2B), which highlights the upregulated and downregulated genes based on log2 fold change and statistical significance (*p*-adj < 0.05). Additionally, a heatmap was generated to display the expression patterns of the DEGs across the samples (Figure 2C), providing a clear visual representation of the clustering of gene expression profiles between the AD and control groups. These results laid the foundation for further investigation into the key modules associated with AD.

### 2.2. Identifying Key Modules Highly Correlated with the Progression of AD via WGCNA

Using WGCNA analysis in GSE132903, we set the soft threshold to 16 at an R^2^ of 0.87 to ensure a biologically meaningful scale-free network (Figure 3A,B). After merging highly correlated modules with a clustering height cutoff of 0.25, a total of 16 modules were detected (Figure 3D). These modified and merged modules are displayed under the cluster tree. Subsequent examination of module correlations revealed no significant associations among them (Figure 3C). The relationship between modules and clinical traits was investigated using the leading eigengene (ME) value’s correlation with clinical features. The sienna3 module was negatively correlated with normal state (r = −0.37, *p* = 1.5 × 10^−7^) and positively correlated with AD (r = 0.37, *p* = 1.5 × 10^−7^), cyan negatively correlated with normal (r = −0.42, *p* = 1.1 × 10^−9^) and positively correlated with AD (r = 0.42, *p* = 1.1 × 10^−9^), orange negatively correlated with normal (r = −0.28, *p* = 7.6 × 10^−5^) and positively correlated with AD (r = 0.28, *p* = 7.6 × 10^−5^), black positively correlated with normal (r = 0.22, *p* = 1.7 × 10^−3^) and negatively correlated with AD (r = −0.22, *p* = 1.7 × 10^−3^), darkgrey positively correlated with normal (r = 0.42, *p* = 1.3 × 10^−9^) and negatively correlated with AD (r = −0.42, *p* = 1.3 × 10^−9^), yellow positively correlated with normal (r = 0.36, *p* = 1.9 × 10^−7^) and negatively correlated with AD (r = −0.36, *p* = 1.9 × 10^−7^), lightcyan positively correlated with normal (r = 0.21, *p* = 2.6 × 10^−3^) and negatively correlated with AD (r = −0.21, *p* = 2.6 × 10^−3^), blue positively correlated with normal (r = 0.43, *p* = 5.7 × 10^−10^) and negatively correlated with AD (r = −0.43, *p* = 5.7 × 10^−10^), darkolivegreen positively correlated with normal (r = 0.28, p = 6.3 × 10^−5^) and negatively correlated with AD (r = −0.28, *p* = 6.3 × 10^−5^), yellowgreen negatively correlated with normal (r = −0.18, *p* = 0.01) and positively correlated with AD (r = 0.18, *p* = 0.01), and grey negatively correlated with normal (r = −0.23, *p* = 1.3 × 10^−3^) and positively correlated with AD (r = 0.23, *p* = 1.3 × 10^−3^) (Figure 3E). Hence, we identified key modules of clinical relevance and further examined all genes within these 11 modules. These key modules lay the foundation for the subsequent analysis in which we investigated the enrichment of immune-related genes within these modules to gain deeper insights into the immune mechanisms underlying AD pathogenesis.

### 2.3. Identifying Overlapping Immune-Related Genes in AD through Intersection Analysis of ImmPort, GEO Database, and WGCNA Modules

To determine the immune genes linked to AD, we obtained 1793 immune-related genes from the ImmPort database. By intersecting these genes with the DEGs retrieved from the GEO database and the pivotal module genes pinpointed in the WGCNA, we uncovered 303 overlapping immune-related genes (Figure 4A). Subsequent enrichment analysis of these overlapping genes revealed that in the GO enrichment analysis, biological processes (BP) were primarily enriched for responses to chemicals, regulation of responses to stimuli, and responses to organic substances. Cellular Components (CC) enrichment was related to the extracellular region, vesicles, and the endomembrane system. Enriched molecular functions (MF) involved signaling receptor binding, molecular function regulator, and signaling receptor activity (Figure 4B and Appendix A). In the KEGG analysis, enrichment was found for pathways related to cancer, cytokine–cytokine receptor interactions, Epstein–Barr virus infection, Influenza A, the MAPK signaling pathway, Herpes simplex virus 1 infection, Kaposi’s sarcoma-associated herpesvirus infection, Human cytomegalovirus infection, antigen processing and presentation, and the PI3K–Akt signaling pathway (Figure 4C and Appendix A). These enrichments underscore the involvement of immune-related processes in AD pathogenesis, highlighting the need to investigate key genes and pathways further.

### 2.4. Machine Learning Identifies Key Genes Linked to AD from Overlapping Immune-Related Genes

To delve deeper into the intricate web of interactions among the overlapping genes we had identified, we constructed a PPI network. This PPI network, comprising 302 nodes and 408 edges, offered a visual representation of the complex gene interactions (Figure 5A). Subsequently, we utilized the cytoHubba plugin in Cytoscape to identify hub genes, employing eight ranking algorithms (BC, EC, CC, DC, LAC, NC, SC, IC). Hub genes are identified through network analysis as genes with high connectivity within a protein–protein interaction (PPI) network. They play central roles in maintaining the integrity and functionality of the network, and their dysregulation can significantly impact various cellular processes. The top 130 genes from each method were selected, and a petal diagram was used to determine 83 common hub genes (Figure 5B). To further identify key genes from these hub genes, we employed two machine learning algorithms: RF and SVM-RFE. Key genes are those that have been identified as crucial for specific biological functions or disease mechanisms, selected based on their functional importance. These genes are often highlighted due to their involvement in critical pathways or significant differential expression in diseased versus normal states. RF is capable of predicting continuous variables and offers predictions without apparent variance, with the advantage of no variable condition restrictions (Figure 5C). SVM-RFE allows for fine screening, focusing on genes with high discriminatory power (Figure 5D,E). A Venn diagram revealed the sweet spot where both methods converged, identifying 25 key genes at their intersection (Figure 5F).

Functional enrichment analysis based on these 25 key genes indicated that in the GO enrichment analysis, BP were primarily enriched for regulation of response to stimulus, positive regulation of response to stimulus, and intracellular signal transduction. CC enrichment related to vesicles, extracellular regions, and the cytoplasm. Enriched MF included enzyme binding, signaling receptor binding, and molecular function regulator (Figure 5G and Appendix A). In the KEGG analysis, enrichment was found for pathways related to the MAPK signaling pathway, measles, hepatitis C, human cytomegalovirus infection, cancer pathways, the TNF signaling pathway, hepatitis B, influenza A, Epstein–Barr virus infection, and human papillomavirus infection, among others (Figure 5H and Appendix A). These findings highlight the crucial roles of these key genes in AD pathogenesis and provide a foundation for the subsequent analysis in which we investigated the expression and validation of these key genes to further elucidate their potential diagnostic and therapeutic implications in AD.

### 2.5. Constructing a Prognostic Model for Alzheimer’s Disease Using Key Genes

Through Venn diagram analysis, we ultimately identified nine upregulated key genes (Figure 6A). Subsequently, we constructed an AD diagnostic nomogram model based on these key genes (*LEP*, *PLXNA3*, *HLA-E*, *DDX58*, *IKBKE*, *VCAM1*, *HSPA6*, *SOS2*, and *HSPA1A*) using the rms package (Figure 6B). This model not only encompassed the complexity of AD but also provided a visual representation for easier understanding and application. To assess the predictive prowess of our model, we generated calibration curves. These curves revealed a negligible disparity between the actual AD risk and the risk predicted by our nomogram, attesting to the model’s remarkable accuracy in forecasting AD (Figure 6C). Furthermore, Decision Curve Analysis (DCA) was employed to evaluate the clinical utility of our model. The “nomogram” curve consistently hovered above the gray line, indicating that our model offers significant benefits to patients—especially those at higher risk thresholds. This further underscores the success of our nomogram construction (Figure 6D).

To validate the diagnostic value of the key genes in AD, we established ROC curves through dataset GSE132903, revealing high diagnostic values for *LEP* (AUC: 0.721229), *PLXNA3* (AUC: 0.742689), *HLA-E* (AUC: 0.691248), *DDX58* (AUC: 0.754103), *IKBKE* (AUC: 0.733537), *VCAM1* (AUC: 0.635283), *HSPA6* (AUC: 0.678098), *SOS2* (AUC: 0.686409), and *HSPA1A* (AUC: 0.615927) in AD (Appendix A). Moreover, in the validation dataset (GSE33000), we quantitatively evaluated the expression levels of key genes in AD. The results demonstrated that expressions of *LEP*, *PLXNA3, HLA-E*, *DDX58*, *IKBKE*, *VCAM1, HSPA6*, *SOS2*, and *HSPA1A* were similarly upregulated in AD tissues (Appendix A), and ROC curve analysis on this validation dataset further corroborated their high diagnostic values for AD (Appendix A). These results validate the diagnostic potential of these key genes in AD, providing a basis for the subsequent analysis in which we explored the expression and screening of these key genes across various cancers using the TCGA database to uncover potential cross-cancer associations and implications for AD treatment strategies.

### 2.6. Utilizing the TCGA Database to Observe the Expression of Key Genes across Cancers

Through extensive pan-cancer analysis utilizing the TCGA database, we systematically explored the expression patterns of key genes identified in AD across various cancer types. This comprehensive study aimed to uncover potential associations and novel insights into the role of these genes in cancer biology. Remarkably, our findings revealed a notable upregulation of several key genes specifically in LIHC. Among these, *PLXNA3*, *HLA-E*, *DDX58*, *IKBKE*, *HSPA6*, *SOS2*, and *HSPA1A* stood out, with significantly elevated expression levels in LIHC compared to other cancer types (Figure 7A–I). This discovery suggests a possible link between these genes and the development or progression of LIHC. To further validate our findings, we conducted a detailed expression analysis in LIHC tissues. Box plots clearly demonstrated that *PLXNA3*, *HLA-E*, *DDX58*, *IKBKE*, *SOS2*, and *HSPA1A* were all significantly upregulated in LIHC tissues compared to normal tissues (Appendix A). Additionally, our paired comparison plots further highlighted the expression differences between normal and LIHC tissues on an individual level. Except for HSPA6, all other key genes (*PLXNA3*, *HLA-E*, *DDX58*, *IKBKE*, *SOS2*, and *HSPA1A*) showed notably higher expression in LIHC samples compared to their normal counterparts. This comparison underscores the specificity and consistency of their upregulation in LIHC (Appendix A). Moreover, to assess the clinical relevance of these findings, we performed survival curve analysis. The results indicated that among the upregulated genes, only *IKBKE* and *HSPA1A* demonstrated a statistically significant association with patient survival. Specifically, high expression levels of these two genes correlated with poorer survival rates in LIHC patients (refer to Appendix A). This suggests that *IKBKE* and *HSPA1A* may serve as valuable prognostic biomarkers for LIHC, aiding in the prediction of disease outcomes and potentially guiding treatment strategies (Appendix A). The pan-cancer analysis revealed intriguing patterns of key gene expression across different cancer types, particularly highlighting their relevance in LIHC. This analysis laid the groundwork for further investigation. Afterwards, we delved into the validation of *IKBKE* and *HSPA1A* expression in LIHC, elucidating their potential role in the tumor microenvironment of LIHC.

### 2.7. Expression and Validation of IKBKE and HSPA1A in LIHC

To validate the expression patterns of *IKBKE* and *HSPA1A* in LIHC, we examined immunohistochemical images obtained from the HPA database. These images revealed a striking increase in the protein expression of both *IKBKE* and *HSPA1A* in LIHC samples when compared to normal liver tissues. This visual evidence strongly supports the upregulation of these genes in cancerous tissues (Figure 8A). Subsequently, we selected other GEO datasets related to LIHC (GSE36376 and GSE39791) as validation groups. ROC curve results demonstrated high diagnostic accuracy for *IKBKE* (AUC = 0.777504 for GSE36376; AUC = 0.643519 for GSE39791) and *HSPA1A* (AUC = 0.779555 for GSE36376; AUC = 0.623746 for GSE39791) in the validation group. These results underscore the potential of these genes as biomarkers for LIHC diagnosis (Figure 8B–E). Box plots further showed that, compared to the normal group, the expression of IKBKE and HSPA1A was significantly upregulated in the LIHC group in the validation datasets (Figure 8F–I). This quantitative data provides additional confirmation of the elevated expression of these genes in cancerous tissues. We further quantitatively analyzed the mRNA levels in human liver cancer tissues using qPCR. The expression of *HSPA1A* and *IKBKE* was significantly elevated in liver cancer tissues, consistent with our previous findings. (Figure 8J,K).

In summary, our comprehensive validation efforts strongly support the upregulated expression of *IKBKE* and *HSPA1A* in LIHC. These findings not only confirm the role of these genes in LIHC but also highlight their potential as diagnostic biomarkers. Moving forward, we delve into the complex relationship between *IKBKE*, *HSPA1A,* and immune infiltration in LIHC, shedding light on the intricate interactions within the tumor microenvironment and paving the way for more targeted therapeutic approaches.

### 2.8. Analysis of Immune Infiltration for IKBKE and HSPA1A in LIHC

To verify the relevance of *IKBKE* and *HSPA1A* in immune cell infiltration in LIHC, we conducted Spearman correlation analysis. The results indicated a positive correlation between *IKBKE* and immune cell infiltration in LIHC (Figure 9A), while *HSPA1A* showed a negative correlation with immune cell infiltration in LIHC (Figure 9B). Digging deeper, lollipop charts illuminated the intricate dynamics at play; IKBKE expression, for instance, was found to be tightly linked to the presence of CD8-positive T cells, indicating a potential role in activating these crucial immune cells. Interestingly, it also showed a negative association with neutrophils, suggesting a complex regulatory role in the immune landscape of LIHC (Figure 9C). On the other hand, *HSPA1A* expressed a significant negative relationship with memory B cells and helper T cells, hinting at its influence in modulating these immune responses (Figure 9D). Therefore, these genes significantly influence the regulation of immune cell infiltration in the LIHC tumor microenvironment.

To further unpack the roles of *IKBKE* and *HSPA1A* in shaping the immune response within LIHC, we leveraged the powerful xCELL algorithm. This analysis provided compelling evidence that both *IKBKE* and *HSPA1A* have the capability to fine-tune the infiltration levels of multiple immune cells in the LIHC environment (Appendix A). Subsequently, the use of the CIBERSORT algorithm and QUANTISEQ algorithm also confirmed that the expression levels of *IKBKE* and *HSPA1A* have a significant impact on infiltrating immune cells (Appendix A). In essence, our comprehensive correlation analysis underscores the profound connection between *IKBKE*, *HSPA1A*, and immune cell infiltration in the LIHC tumor microenvironment. This exploration not only sheds light on the complex immune dynamics within LIHC but also lays the groundwork for future investigations. We are eager to delve deeper into the single-cell expression patterns of *IKBKE* and *HSPA1A* in LIHC, with the ultimate goal of unraveling their precise functions within distinct immune cell subpopulations of the tumor microenvironment.

### 2.9. Expression of IKBKE and HSPA1A at the Single-Cell Level in LIHC

In previous studies, the analysis of gene expression in LIHC primarily focused on tissue samples and cellular populations. To gain a more precise understanding of the expression patterns of *IKBKE* and *HSPA1A* in different immune cell and cancer cell subpopulations, as well as their potential roles within the tumor microenvironment, we explored their expression in LIHC using single-cell transcriptomic analysis. In LIHC, we conducted an in-depth single-cell transcriptomic analysis of *IKBKE* and *HSPA1A* using four datasets from the TISCH database (LIHC-GSE125449, LIHC-GSE140228 10X, LIHC-GSE140228 Smartseq2, LIHC-GSE98638). Heatmaps revealed significant upregulation of *IKBKE* in CD4Tconv, Treg, Tprolif, CD8T, and CD8Tex cells. This upregulation suggests a potential role for *IKBKE* in the activation, proliferation, or regulatory functions of these immune cell subsets within the LIHC tumor microenvironment (Figure 10A). *HSPA1A* showed significant upregulation in Treg, CD8Tex, B, Plasma, Mono/Macro, Endothelial, and Fibroblasts cells. This finding indicates that *HSPA1A* might play a crucial role in the stress response, immune modulation, or tissue remodeling within the tumor (Figure 10B). To further elaborate on these expression patterns, we utilized t-SNE plots to visualize the intricate distribution of *IKBKE* and *HSPA1A* expression across various immune cell subpopulations in LIHC. These plots not only confirm the upregulation observed in the heatmaps but also reveal the heterogeneity of expression within each cell type, shedding light on the complexity of the immune response and tumor microenvironment in LIHC (Figure 10C–E). Collectively, our findings at the single-cell level strongly implicate *IKBKE* and *HSPA1A* in the regulation of local immune responses and the modulation of the tumor microenvironment in LIHC. The observed upregulation patterns suggest potential targets for immunotherapy and highlight the importance of considering cellular heterogeneity when designing treatment strategies for LIHC.

## 3. Discussion

Cancer and AD are two prevalent disorders whose ultimate pathophysiological mechanisms remain unclear. In our analysis of the GSE132903 dataset, we conducted a thorough examination of gene expression profiles to identify 14,677 DEGs associated with AD. Through WGCNA, we further elucidated key module genes that are closely linked to the pathogenesis of AD. By integrating these DEGs, module genes, and leveraging data from the ImmPort database, we employed PPI network analysis to pinpoint hub genes crucial in AD. To further refine our identification of key genes within these hub genes, we utilized two advanced machine learning algorithms: RF and SVM-RFE. These methods allowed us to prioritize genes based on their importance and predictive power within the dataset [24]. By combining the results of RF and SVM-RFE, we identified nine key genes consistently highlighted for elevated expression in AD: LEP, *PLXNA3*, *HLA-E*, *DDX58*, *IKBKE, VCAM1*, *HSPA6*, *SOS2*, and *HSPA1A*. To validate the significance of these genes in LIHC, we analyzed LIHC-related datasets. This validation process confirmed the significant differential expression of *IKBKE* and *HSPA1A* in LIHC tissues. Furthermore, our q-PCR results revealed a substantial increase in the transcription levels of *IKBKE* and *HSPA1A* in LIHC tumor tissues compared to adjacent normal tissues. This suggests a potential role for these genes in the development and progression of LIHC. In-depth immune infiltration and single-cell analyses were then conducted to understand how *IKBKE* and *HSPA1A* might influence the immune microenvironment in LIHC. These analyses unveiled the potential of these genes to modulate the immune response in LIHC, thereby impacting disease progression. Taken together, these findings indicate that *IKBKE* and *HSPA1A* may serve as crucial factors in the onset and development of both AD and LIHC, highlighting them as promising targets for future therapeutic strategies.

Inflammation, a pivotal factor in AD and LIHC, underscores the intricate interplay between chronic inflammation and neurodegeneration, as well as cancer development [25]. The relationship between inflammation and cancer is well-documented, with LIHC often arising from a background of chronic inflammation and fibrosis [26,27]. Our bioinformatics analysis uncovered that *IKBKE* and *HSPA1A* play pivotal roles in modulating immune cell infiltration in AD and LIHC patients, thereby reshaping the immune microenvironment. *IKBKE*, a key activator of the NF-κB pathway, is instrumental in regulating NF-κB-mediated inflammatory responses, immune cell activation, and tumorigenesis [28]. This is achieved through the phosphorylation of IκB, an inhibitor of NF-κB, leading to IκB’s degradation and subsequent release of NF-κB into the nucleus to initiate downstream gene expression [29,30,31]. Such regulation is crucial for immune responses, inflammation, and in some instances, tumor initiation and progression. *HSPA1A*, a member of the Heat Shock Protein (HSP) family [32], is implicated in exacerbating inflammation by stimulating the secretion of pro-inflammatory cytokines [33,34]. These findings underscore the pivotal roles of *IKBKE* and *HSPA1A* in modulating the pathogenesis of AD and LIHC, thus highlighting their potential as promising therapeutic targets. Targeting these pathways may offer novel therapeutic strategies for managing AD and LIHC by modulating the inflammatory microenvironment.

In this study, we integrated several advanced techniques, including WGCNA, PPI, and machine learning algorithms, to identify the key genes *IKBKE* and *HSPA1A* and assess their diagnostic value for patients with AD and LIHC. Furthermore, validation through multiple independent datasets and experiments enhanced the credibility and applicability of our findings. Nevertheless, a pivotal limitation of the present study lies in the absence of laboratory validation pertaining to associated signaling pathways. Hence, future endeavors will necessitate the experimental validation of pertinent signaling pathways, aiming to further elucidate the functional roles of *IKBKE* and *HSPA1A* and to investigate their specific implications in the therapeutic management of AD and LIHC.

## 4. Materials and Methods

### 4.1. Data Collection and Differential Gene Analysis

We retrieved four datasets from the Gene Expression Omnibus (GEO) database using “AD” and “LIHC” as keywords, including AD datasets GSE132903 and GSE33000 and LIHC datasets GSE36376 and GSE39791 [35]. GSE132903 served as the primary dataset for analysis, while GSE33000, GSE36376, and GSE39791 were utilized as validation sets for further verification (Appendix A). Subsequently, differential expression genes (DEGs) analysis related to AD was conducted on the GSE132903 dataset using the “limma” package (Available online: https://www.bioconductor.org/packages/release/bioc/html/limma.html (accessed on 13 November 2023), within the R environment, version 4.3.0), and volcano plots were generated [36]. The criteria for significant differential expression were defined as a log2|fold change (FC)| > 1 and a *p*-value < 0.05. Finally, heatmaps of the selected DEGs were produced using the pheatmap package (Available online: https://cran.r-project.org/web/packages/pheatmap/index.html (accessed on 13 November 2023), within the R environment, version 4.3.0). The purpose of this analysis was to identify key genes associated with AD, providing a foundation for subsequent functional and pathway analyses.

### 4.2. Identification of Key Gene Modules Associated with AD via WGCNA

Using the “WGCNA” package (Available online: https://cran.r-project.org/web/packages/WGCNA/index.html (accessed on 12 January 2024), within the R environment, version 4.3.0), we conducted Weighted Gene Co-expression Network Analysis (WGCNA) to identify key modules associated with AD in dataset GSE132903. This analysis aimed to uncover gene relationships through co-expression networks, thereby enhancing our understanding of the molecular mechanisms underlying the disease. Initially, samples were clustered to identify potential outliers. Subsequent automated network construction facilitated the formation of a co-expression network. Gene modules were delineated through hierarchical clustering combined with dynamic tree cutting. To associate these modules with clinical traits, we assessed module membership (MM) and gene significance (GS). Modules exhibiting an absolute threshold of Pearson MM correlation and a p-value below 0.05 were designated as central modules. Criteria of MM > 0.8 and GS > 0.2 indicated a significant association with clinical attributes. For these significant modules, in-depth gene-specific analysis was performed [37].

### 4.3. Enrichment Analysis of DEGs

Utilizing the Metascape online platform (https://metascape.org/ accessed on 14 January 2024) [38], we conducted a comprehensive enrichment analysis encompassing Gene Ontology (GO) annotations—including Biological Processes (BP), Cellular Components (CC), and Molecular Functions (MF)—as well as KEGG pathways. This analysis was crucial for understanding the functional roles and biological processes in which DEGs are involved. Annotations and pathways with an adjusted *p*-value below 0.05 were considered significantly enriched.

### 4.4. Construction of the PPI Network and Identification of Hub Genes

We constructed a Protein–Protein Interaction (PPI) network using the STRING database, setting a stringent interaction score threshold of >0.9 [39]. This analysis aimed to identify potential interactions between proteins encoded by DEGs that could provide insights into their functional roles. For enhanced visualization, we employed Cytoscape software (version 3.9.1) [40]. Within Cytoscape, the CytoHubba plugin facilitated the identification of pivotal genes [41]. We selected eight distinct algorithms from CytoHubba’s ranking methodologies, namely: Betweenness Centrality (BC), Eigenvector Centrality (EC), Closeness Centrality (CC), Degree Centrality (DC), Local Average Connectivity-based method (LAC), Network Centrality (NC), Subgraph Centrality (SC), and Information Centrality (IC). Aggregating results across these eight approaches, we chose the top 130 genes from each method and utilized a Venn diagram to determine the hub genes common across all algorithms (Appendix A). This comprehensive approach ensured the robust identification of key genes involved in disease progression.

### 4.5. Machine Learning and Identification of Key Genes

In this study, we employed advanced machine learning techniques, including Random Forest (RF) [42,43,44] and Support Vector Machine Recursive Feature Elimination (SVM-RFE) [45,46], to identify key genes associated with AD. These machine learning methods are powerful tools for selecting the most informative features from large datasets, enhancing the predictive power of our models. The RF method systematically ranked genes, with those exhibiting a relative importance exceeding 0.25 identified as significant determinants. The SVM-RFE technique iteratively trained various feature subsets to pinpoint the most influential predictors. Rigorous ten-fold cross-validation was utilized to optimize parameters, ensuring the minimization of partial likelihood bias. Genes at the intersection of RF and SVM-RFE were deemed critical for the diagnosis of AD.

### 4.6. Predictive Modeling of AD Recurrence and Evaluation of Key Genes

Following the identification of key genes associated with AD, we meticulously constructed a composite nomogram model using the “rms” package (Available online: https://cran.r-project.org/web/packages/rms/index.html (accessed on 20 January 2024), within the R environment, version 4.3.0), aimed at predicting the recurrence of AD. This model provided a visual representation of how different clinical factors contribute to recurrence risk. In this model, “Scores” represent the weighted values of individual clinical factors, while “Total Score” encompasses the cumulative weights of all considered factors. The reliability and predictive capability of our model were rigorously assessed via calibration curves. To ensure the clinical applicability of our model, we employed decision curve analysis to scrutinize its clinical impact curves. Additionally, we generated a Receiver Operating Characteristic (ROC) curve using the pROC function in R, with the Area Under the Curve (AUC) serving as a metric to evaluate the diagnostic utility of the key genes [47].

### 4.7. Expression and Analysis of Key Genes in Hepatocellular Carcinoma

We downloaded clinical data for LIHC from The Cancer Genome Atlas database (https://tcga-data.nci.nih.gov/tcga/ accessed on 11 February 2024). This step was essential for validating the relevance of key genes identified in AD within the context of another disease. The expression of key genes was compared between case and control groups to identify any significant differences. Furthermore, the prognostic relevance for patients with LIHC was evaluated using the Kaplan–Meier plotter tool (Available online: https://kmplot.com/analysis/ (accessed on 11 February 2024)). Lastly, immunohistochemical images of key genes in LIHC and normal tissues were sourced from the Human Protein Atlas (HPA) database (https://www.proteinatlas.org/ accessed on 13 February 2024). These images were utilized to assess the differential expression of key genes at the protein level.

### 4.8. Immune Infiltration and Single-Cell Analysis

We explored the relationship between the expression of key genes and immune infiltration in LIHC using algorithms such as TIMER, EPIC, IPS, MCP-counter, xCELL, CIBERSORT, and QUANTISEQ. This analysis aimed to elucidate the role of key genes in the tumor immune microenvironment, providing insights into potential immunotherapeutic targets. A *p*-value < 0.05 was considered statistically significant for immune cell infiltration [48]. Subsequently, utilizing the TISCH database [49], we delved into the single-cell expression landscape of key genes within the microenvironment of LIHC.

### 4.9. Quantitative Real-Time RT-PCR

Our study encompassed a total of 3 adult inpatients diagnosed with LIHC. The research protocol adhered strictly to the principles outlined in the Declaration of Helsinki, ensuring the ethical treatment of human tissues. Additionally, the study received approval from the Clinical Research Ethics Committee of Hebei Medical University. Prior to their participation, each patient provided informed consent by signing the necessary forms. The total RNA from the human liver tissues was extracted using the RNA isolation kit (RNAiso, Takara, San Jose, CA, USA). In this experiment, the isolated RNA was dissolved in 20 mL of DEPC-treated water and then reverse-transcribed using a reverse transcription reagent kit (PrimeScript RT Reagent Kit with gDNA Eraser, Takara) and a thermal cycler (Mastercycler, Eppendorf, Hamburg, Germany). The cDNA was used for the qPCR assay and the amplification curves were produced using SYBR Premix Ex TaqII from Takara. This experimental validation ensures the accuracy and reliability of the identified gene expression levels, reinforcing the robustness of our results. Primers used in the quantitative PCR were: GAPDH: forward-AATGGACAACTGGTCGTGGAC; reverse-CCCTCCAGGGGATCTGTTTG; *HSPA1A*: forward-CATCGCCTATGGGCTGGAC; reverse-GGAGAGAACCGACACATCGAA; *IKBKE*: forward-GAGGGGCAGAGCATCTACAAG; reverse-AGACCGAGACGAACTTCTCATC.

### 4.10. Statistical Analysis

The Wilcoxon rank-sum test was utilized to assess statistical significance between two groups. Correlation analysis was performed using the Spearman correlation coefficient. All statistical analyses were conducted using R software (version 3.6.3), with a two-tailed *p*-value < 0.05 considered statistically significant.

## 5. Conclusions

This study, by integrating bioinformatics analysis with machine learning techniques, has identified that *IKBKE* and *HSPA1A* exert regulatory effects on the incidence and progression of both Alzheimer’s Disease (AD) and Liver Hepatocellular Carcinoma (LIHC) through the modulation of immune cell infiltration. This provides new strategies for the diagnosis and treatment of these two diseases in the future, and lays a solid theoretical foundation for further exploration into AD and LIHC.

## Figures and Tables

**Figure 1 ijms-25-06934-f001:**
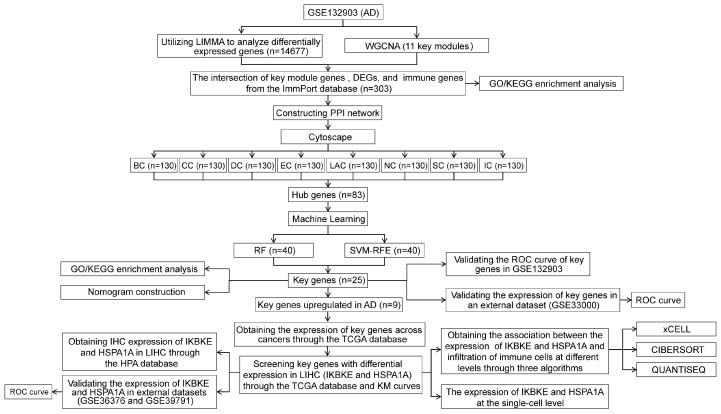
Flow chart of the study, starting with the analysis of DEGs from GSE132903 using Linear Models for Microarray Data (LIMMA) and identifying key gene modules with WGCNA. A PPI network was constructed and hub genes were identified using CytoHubba. RF and SVM-RFE were employed to pinpoint key genes, which were validated in external datasets. Their relevance in LIHC was examined, including their expression, prognostic value, and relationship with immune infiltration. Alzheimer’s Disease (AD), Liver Hepatocellular Carcinoma (LIHC), Differentially Expressed Genes (DEGs), Linear Models for Microarray Data (LIMMA), Weighted Gene Co-expression Network Analysis (WGCNA), Protein-protein Interaction (PPI), Betweenness Centrality (BC), Eigenvector Centrality (EC), Closeness Centrality (CC), Degree Centrality (DC), Local Average Connectivity-based method (LAC), Network Centrality (NC), Subgraph Centrality (SC), and Information Centrality (IC), Random Forest (RF), Support Vector Machine Recursive Feature Elimination (SVM-RFE), The Cancer Genome Atlas (TCGA), Human Protein Atlas (HPA), Gene Ontology (GO), Kyoto Encyclopedia of Genes and Genomes (KEGG), Kaplan-Meier (KM), Receiver Operating Characteristic (ROC).

**Figure 2 ijms-25-06934-f002:**
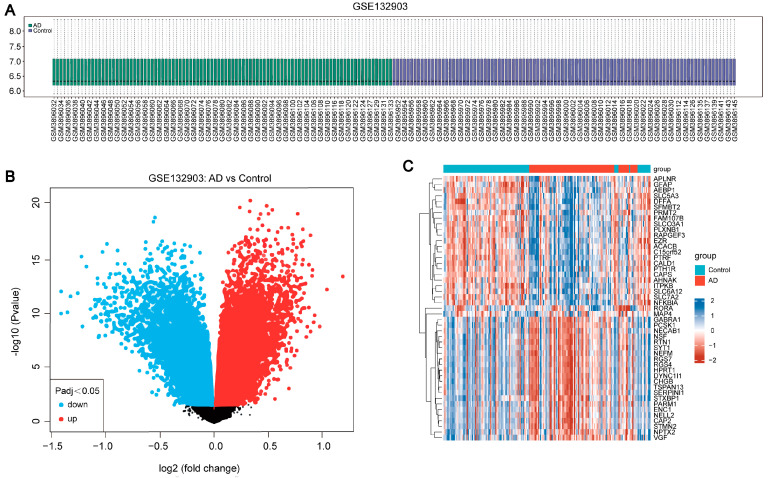
Differential Gene Expression Analysis in AD versus Control Samples in GSE132903. (**A**) Box plot shows the distribution of normalized expression values for each sample, ensuring data quality and comparability. (**B**) Volcano plot illustrates the differentially expressed genes (DEGs), with upregulated genes in red and downregulated genes in blue, based on log2 fold change and statistical significance (*p*-adj < 0.05). (**C**) Heatmap depicts the expression patterns of DEGs across AD and control samples, with hierarchical clustering applied to both samples and genes, highlighting distinct expression profiles.

**Figure 3 ijms-25-06934-f003:**
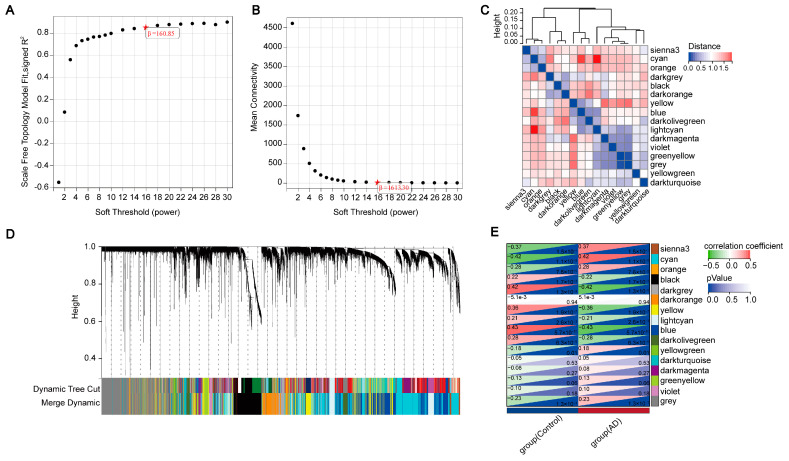
Construction of the WGCNA. (**A**,**B**) The soft-thresholding power was set to 16, achieving a scale-free topology fit index (R^2^) of 0.87. (**C**) Heatmap of collinearity for module eigengenes. (**D**) Dendrogram of original and merged modules, with a total of 16 modules detected after merging highly correlated modules with a clustering height cutoff of 0.25. (**E**) Heatmap of module-trait relationships, indicating the correlation between module eigengenes (MEs) and clinical traits, such as normal state and AD (Alzheimer’s Disease).

**Figure 4 ijms-25-06934-f004:**
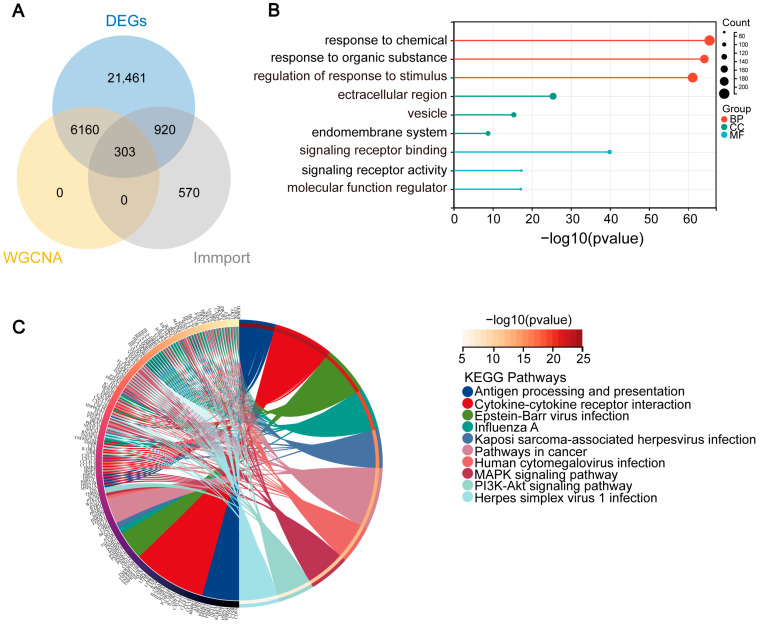
Enrichment Analysis of Immune-Related Overlapping Genes. (**A**) Venn diagram showing the overlap of immune-related genes from the ImmPort database, key module genes from WGCNA, and DEGs from the GEO database. (**B**) Gene Ontology (GO) analysis of the overlapping genes, with categories for biological processes (BP), cellular components (CC), and molecular functions (MF). (**C**) KEGG pathway analysis of the overlapping genes, indicating the pathways involved in immune-related processes.

**Figure 5 ijms-25-06934-f005:**
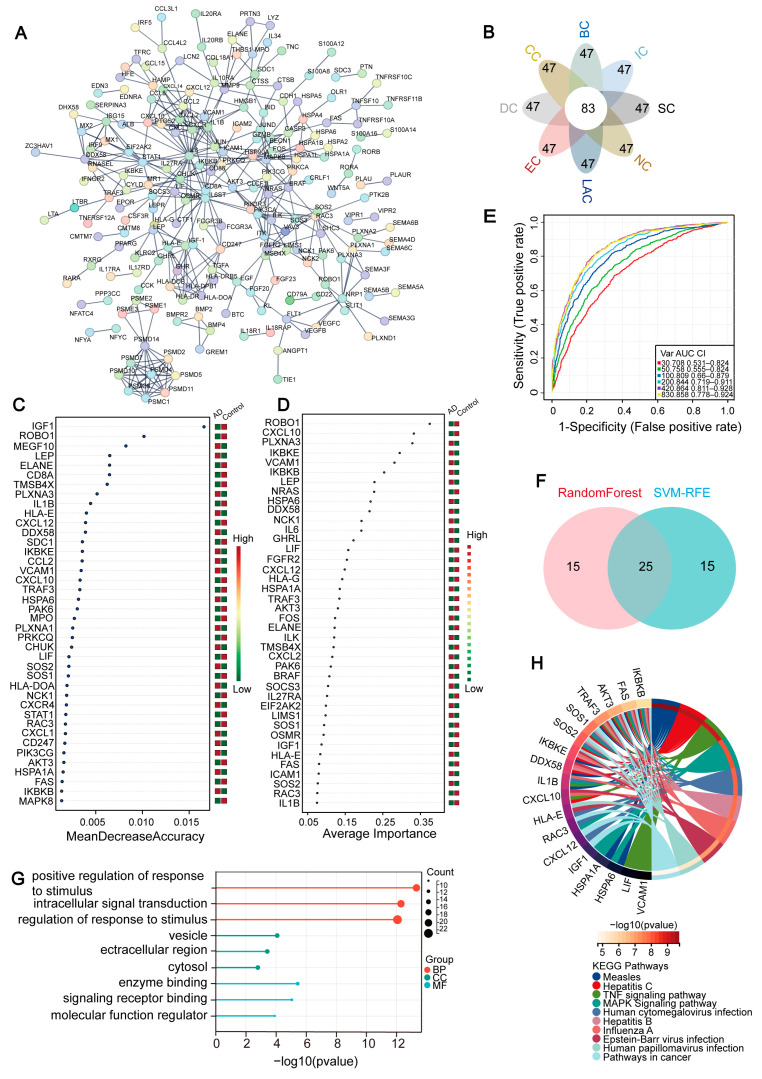
Identification and Enrichment Analysis of Key Genes. (**A**) The PPI network diagram visually demonstrates the interconnectedness of genes, aiding in the understanding of which genes might collaborate in biological processes. (**B**) The Venn diagram reveals the overlap of hub genes pinpointed through PPI analysis using eight distinct ranking algorithms, highlighting genes recognized as significant nodes across multiple methods. (**C**) Critical genes are further honed using the RF algorithm, selecting those with a high importance score within the model. (**D**,**E**) The SVM-RFE algorithm is employed to iteratively eliminate less discriminatory genes, ultimately identifying the most influential ones. (**F**) A Venn diagram illustrates the convergence of key genes pinpointed by both the RF and SVM-RFE techniques, emphasizing their shared importance. (**G**) GO enrichment analysis offers insights into the primary biological functions, cellular components, and molecular processes enriched among the selected genes. (**H**) Associated KEGG pathways are presented, delving into the genes’ potential roles in cell metabolism, signal transduction, and other vital biological processes.

**Figure 6 ijms-25-06934-f006:**
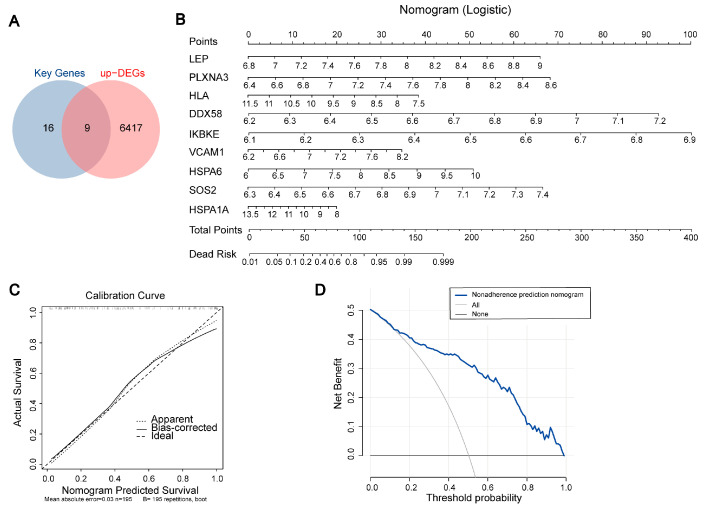
Construction and Validation of the AD Diagnostic Nomogram Model. (**A**) Venn diagram showing the overlap between key genes and upregulated DEGs. (**B**) Predicting the occurrence of AD using a nomogram. (**C**) A calibration curve is presented to evaluate the predictive accuracy of the nomogram model, demonstrating its reliability in forecasting AD occurrence. (**D**) DCA is employed to assess the clinical utility and net benefit of the nomogram model in the context of AD diagnosis.

**Figure 7 ijms-25-06934-f007:**
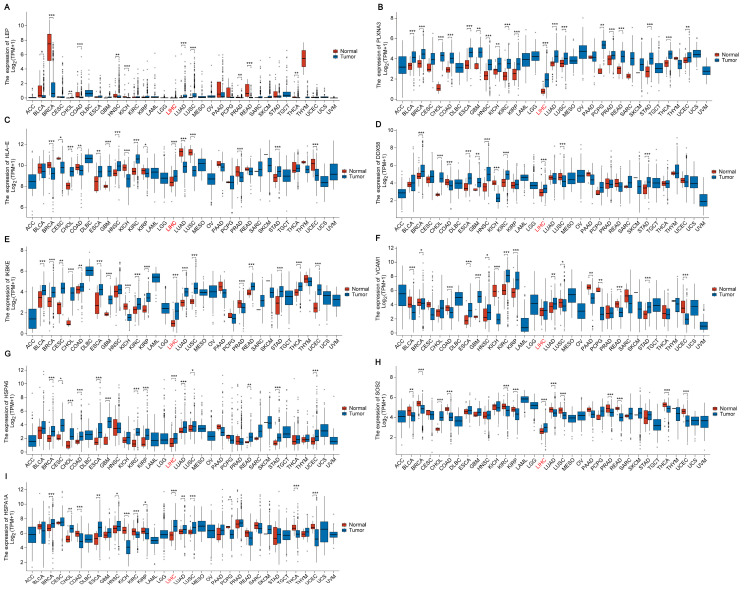
Expression of Key Genes Across Pan-Cancer Types: (**A**) *LEP*. (**B**) *PLXNA3*. (**C**) *HLA-E.* (**D**) *DDX58*. (**E**) *IKBKE*. (**F**) *VCAM1*. (**G**) *HSPA6*. (**H**) *SOS2*. (**I**) *HSPA1A*. The LIHC group is highlighted in red. *, ** and *** indicate significant difference from the Normal group, * *p* < 0.05, ** *p* < 0.01, *** *p* < 0.001 compared with Normal group.

**Figure 8 ijms-25-06934-f008:**
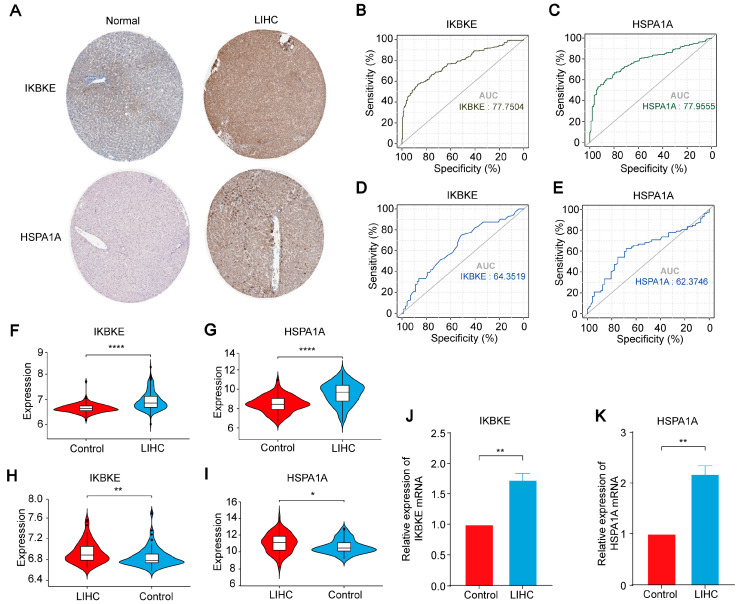
Expression and Validation of *IKBKE* and *HSPA1A* in LIHC. (**A**) Immunohistochemical staining reveals the expression patterns of *IKBKE* and *HSPA1A* in normal liver tissues compared to LIHC tissues. (**B**,**C**) ROC curve analysis demonstrates the diagnostic accuracy of *IKBKE* and *HSPA1A* in the validation dataset GSE36376. (**D**,**E**) ROC curve analysis further validates the diagnostic utility of *IKBKE* and *HSPA1A* in another dataset, GSE39791. (**F**,**G**) Quantitative expression analysis confirms the differential expression of *IKBKE* and *HSPA1A* in the GSE36376 dataset. (**H**,**I**) Similar expression patterns of *IKBKE* and *HSPA1A* are observed in the GSE39791 dataset. (**J**,**K**) Quantitative analysis of *IKBKE* and *HSPA1A* mRNA transcription levels in LIHC patients (*n* = 3) versus adjacent non-cancerous tissue samples (*n* = 3) provides further evidence of their differential expression in vivo. *, ** and **** indicate significant difference from the control group, * *p* < 0.05, ** *p* < 0.01, **** *p* < 0.0001 compared with control group.

**Figure 9 ijms-25-06934-f009:**
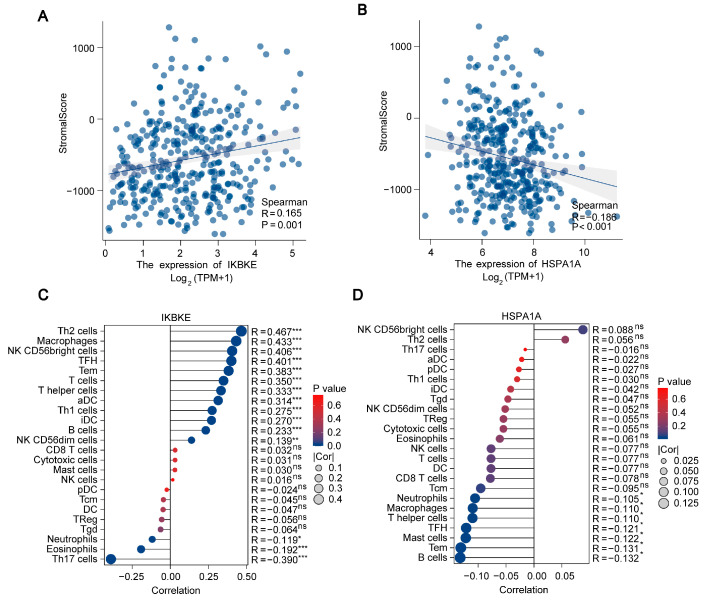
Analysis of Immune Infiltration for *IKBKE* and *HSPA1A* in LIHC. (**A**,**B**) Spearman correlation analysis graphs illustrate the relationship between the expression levels of *IKBKE* and *HSPA1A*, respectively, and the degree of immune cell infiltration in LIHC. (**C**,**D**) Lollipop charts further detail the correlation between the expression levels of *IKBKE* and *HSPA1A*, respectively, and various immune cell types. *, ** and *** indicate significant difference from the control group, ^ns^ indicates no significant difference, * *p* < 0.05, ** *p* < 0.01, *** *p* < 0.001 compared with control group.

**Figure 10 ijms-25-06934-f010:**
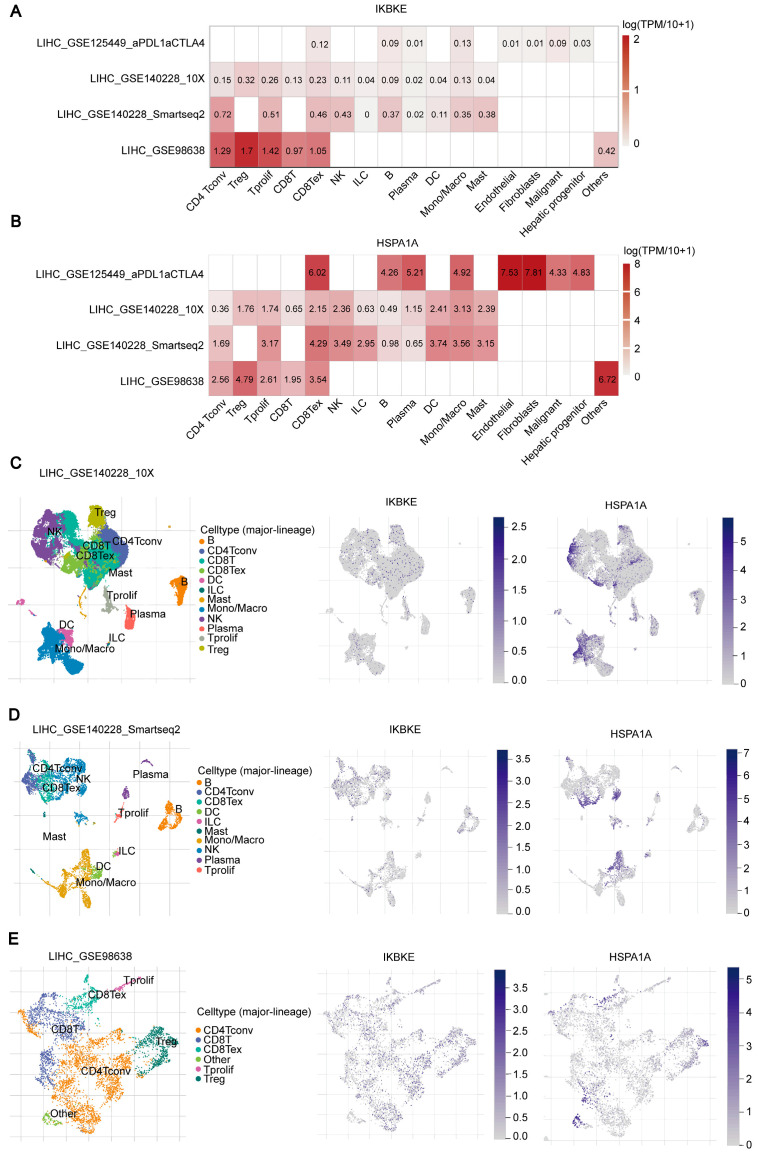
Single-Cell RNA Sequencing Analysis of *IKBKE* and *HSPA1A* in LIHC. (**A**,**B**) Heatmaps reveal the distinct expression patterns of *IKBKE* and *HSPA1A* across a diverse range of immune cell types within the LIHC datasets, highlighting their potential roles in immune cell differentiation and function. (**C**–**E**) The t-SNE plots further visualize the intricate distribution of *IKBKE* and *HSPA1A* expression within the immune cell population of LIHC, offering insights into their spatial and functional heterogeneity within the tumor microenvironment.

## Data Availability

All data in this study are included in this published article.

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
