# Peer review of "Exploring the Roles of Key Mediators IKBKE and HSPA1A in Alzheimer’s Disease and Hepatocellular Carcinoma through Bioinformatics Analysis"

_ijms, 2024, doi:10.3390/ijms25136934_

Round 1

Reviewer 1 Report

Comments and Suggestions for Authors

This study explores a link between Alzheimer's Disease (AD) and Liver Hepatocellular Carcinoma (LIHC) by identifying shared key genes that could serve as diagnostic biomarkers and therapeutic targets. Through various analytical methods, the researchers identified nine key genes in AD, with IKBKE and HSPA1A also relevant for LIHC. These genes may influence disease progression by modulating immune cell infiltration, providing a basis for targeted therapies. Below are some major and minor comments:

Line 18: Mention the full form of ROC in the abstract before using the abbreviation.

Figure 1: The legend does not clearly describe the figure. The abbreviations are not mentioned in the legend, making it difficult to understand the complex work. Provide clear explanations and define all abbreviations.

Line 57: The authors should mention the full forms of abbreviations the first time they appear, such as DEGs, which stands for differentially expressed genes.

Lines 61-62: The results related to Figure 2 are not well described. The figure legend is poorly detailed.

Justification for Analyses: The authors have not justified the reasons for conducting multiple analyses. Providing a rationale for each analysis would strengthen the study's credibility.

Connectivity: There is a lack of connectivity between the results. Ensure that the findings are logically connected and flow coherently throughout the manuscript.

Figure Descriptions: The figures, although visually appealing, are not well described in the main text or the legends. Improve the descriptions to enhance understanding.

Discussion of Results: The results are poorly discussed. A more thorough discussion is needed to contextualize the findings within the existing literature and highlight their significance.

Supplementary information: There is no need to include all the data in the main manuscript, especially using small figures that are not visible.

Manuscript Revision: The manuscript requires significant revisions to address these issues. Once revised, the manuscript has the potential for good readership and novelty.

Overall, while the study presents promising findings, it requires substantial improvement.

Comments on the Quality of English Language

Moderate editing of English language required

Reviewer 2 Report

Comments and Suggestions for Authors

Alzheimer’s disease (AD) is a key research area. The authors have presented an interesting work on finding the link between AD and cancer. The study aimed to find the genes common to both the diseases. Such information will provide information about the underlying disease mechanism. Below are some minor questions:

1.     What is the difference between Hub genes and key genes mentioned in the paper? The authors can add a brief note when they are mentioned for the first time in the manuscript.

2.     How did the authors select the specific machine learning techniques? How are these different from other machine learning techniques?

Round 2

Reviewer 1 Report

Comments and Suggestions for Authors

Thank you for including the suggestions and comments. However, there is a minor change required that the author should discuss the results of the random forest and support vector machine and include the following reference:

https://www.nature.com/articles/s41598-018-33476-x
